# All Eyes on the Prefusion-Stabilized F Construct, but Are We Missing the Potential of Alternative Targets for Respiratory Syncytial Virus Vaccine Design?

**DOI:** 10.3390/vaccines12010097

**Published:** 2024-01-18

**Authors:** Sofie Schaerlaekens, Lotte Jacobs, Kim Stobbelaar, Paul Cos, Peter Delputte

**Affiliations:** 1Laboratory for Microbiology, Parasitology and Hygiene, University of Antwerp (UA), Universiteitsplein 1 S.7, 2610 Antwerp, Belgium; sofie.schaerlaekens@uantwerpen.be (S.S.); lotte.jacobs@uantwerpen.be (L.J.); kim.stobbelaar@uantwerpen.be (K.S.); paul.cos@uantwerpen.be (P.C.); 2Pediatrics Department, Antwerp University Hospital (UZA), Wilrijkstraat 10, 2650 Edegem, Belgium; 3Infla-Med Centre of Excellence, University of Antwerp (UA), Universiteitsplein 1 S.7, 2610 Antwerp, Belgium

**Keywords:** RSV, vaccine

## Abstract

Respiratory Syncytial Virus (RSV) poses a significant global health concern as a major cause of lower respiratory tract infections (LRTIs). Over the last few years, substantial efforts have been directed towards developing vaccines and therapeutics to combat RSV, leading to a diverse landscape of vaccine candidates. Notably, two vaccines targeting the elderly and the first maternal vaccine have recently been approved. The majority of the vaccines and vaccine candidates rely solely on a prefusion-stabilized conformation known for its highly neutralizing epitopes. Although, so far, this antigen design appears to be successful for the elderly, our current understanding remains incomplete, requiring further improvement and refinement in this field. Pediatric vaccines still have a long journey ahead, and we must ensure that vaccines currently entering the market do not lose efficacy due to the emergence of mutations in RSV’s circulating strains. This review will provide an overview of the current status of vaccine designs and what to focus on in the future. Further research into antigen design is essential, including the exploration of the potential of alternative RSV proteins to address these challenges and pave the way for the development of novel and effective vaccines, especially in the pediatric population.

## 1. Introduction

Respiratory Syncytial Virus (RSV) was first identified in 1956 and is one of the leading causes of lower respiratory tract infections (LRTIs) worldwide [1,2]. RSV is a member of the *Pneumoviridae* family, associated with the genus *Orthopneumovirus*, and was recently reclassified by ICTV to the species *Human Orthopneumovirus* (hOPV) [3]. It can be divided into two major antigenic subtypes, RSV-A and RSV-B, based on the sequence of the G protein. These two subtypes can cocirculate, with predominance alternating every 1–2 years [4,5].

Every year, there are around 101.400 deaths among children under the age of 5 due to RSV infection [6]. Nearly half of the children have contracted RSV within the first year of life, and by the age of 2, almost all of them will have been infected at least once [7,8]. In most children, mild symptoms such as a runny nose, cough and/or decreased appetite are observed. However, more serious infections can result in hospitalization with pneumonia or bronchitis, with a peak between the ages of 2 and 4 months old [8,9]. An estimated 33 million cases, with 3.6 million hospitalizations are registered on a yearly basis in children under the age of 5 [6,10]. The main concern is located in low- and middle-income countries (LMICs), where more than 97% of RSV-related deaths occur. In high-income countries (HICs), fatal cases are often premature infants or infants with underlying conditions such as chronic lung or congenital heart disease, Down’s syndrome and neuromuscular disorders [6,11,12,13,14,15].

Throughout life, people are repeatedly reinfected with RSV, as RSV does not confer long-lasting immunity [16]. In healthy adults, the disease often manifests with mild symptoms restricted to the upper airway tract. However, severe disease can develop in immunocompromised adults and the elderly (65+). In these populations, RSV can be a significant cause of severe respiratory illness and a major cause of mortality, with mortality rates approaching those of seasonal influenza [17,18,19,20,21]. The Center for Disease Control (CDC) estimates that each year, RSV hospitalizations among older persons in the United States range from 60,000 to 160,000, and deaths range from 6000 to 10,000. Another study covering several HICs suggests the annual incidence of RSV in the elderly is approximately 5.2 million. Among these cases, 470,000 individuals require hospitalization, while the mortality rate stands at 33,000 [9,22,23,24,25,26].

Overall, RSV infections represent a substantial public health concern, particularly in young children, the elderly and vulnerable populations. Further research and interventions are necessary to reduce the burden of RSV-associated illnesses and mortality. While vaccines are now available for the elderly and newborns through maternal vaccination, prophylaxis for infants older than 6 months is limited to passive antibody administration. Further research into the mechanisms of protection in infants and the development of novel vaccines is necessary to reduce the burden of RSV-associated illnesses and mortality in infants.

## 2. RSV Structure

The RSV genome consists of a linear, negative-sense, single-stranded RNA molecule of around 15 kb, comprising 10 genes that encode 11 structural and non-structural proteins (see Figure 1B) [27]. The viral RNA is surrounded by the ribonucleoprotein complex (RNP), which consists of three structural proteins: the nucleoprotein (N), the phosphoprotein (P) and the large protein (L) [28]. The matrix (M) protein is a non-glycosylated phosphorylated protein that acts as a bridge between the lipid bilayer envelope and the nucleocapsid (NC) and, therefore, plays a role in viral assembly [29,30,31,32]. Two overlapping open reading frames (ORFs) give rise to the transcription processivity factor M2-1 and the transcriptional regulatory protein M2-2 [33]. Non-structural proteins NS1 and NS2 are known to interfere with antiviral immune responses by means of altering signaling pathways in the interferon-mediated signaling cascade and cytokine gene expression [34,35]. Three main viral proteins are displayed on the virus envelope: the small hydrophobic (SH) protein, the attachment protein (G) and the fusion (F) protein (see Figure 1A) [36].

The SH protein facilitates membrane permeabilization by forming ion channels [37]. The G protein is responsible for attaching the virion to the respiratory epithelial cells and immune cells via interaction with target cell receptors [38]. The F protein mediates fusion of the viral membrane with the host cell membrane and was shown to interact with nucleolin [39]. Furthermore, F expressed on the surface of the infected cells also mediates fusion with adjacent cells, resulting in the formation of multinucleated cells called syncytia [40,41,42].

The F protein is initially transcribed as an inactive precursor, F0 [43]. To become fusion-competent, F needs to be cleaved by furin-like proteases. A unique property of RSV is that the F protein does not contain one but two furin cleavage sites (RARR109 and KKRKRR136) [44]. This dual cleavage results in the removal of a 27-aa peptide (p27) and generates the heterodimeric protomer comprising a fragment F2 and a fragment F1, covalently linked through two disulfide bonds (Figure 2) [45,46].

For many years, it was believed that the removal of p27 through furin cleavage was completed before RSV F trimerization, cell surface expression and its incorporation into virus particles [44]. Consequently, p27 would not be exposed to the immune system and was considered not essential for vaccine antigen design [48]. However, an increasing number of research groups are now challenging the hypothesis that double cleavage is essential for surface expression and virion incorporation [49]. The precise function of p27 in the context of anti-RSV immunity remains unclear, necessitating further investigation to assess its significance in vaccine development.

The F protein contains five to six N-linked glycosylation sites, of which two to three sites are located in the p27 peptide, depending on the virus isolate [47]. The mature F is formed by the trimerization of the protomers. The mature F protein is expressed on the virion or cell membrane as a metastable prefusion conformation that can readily rearrange into the lower-energy postfusion conformation (Figure 3) [38]. Recent evidence suggests the trimeric prefusion F is not a locked structure but rather a flexible one, balancing temporary trimer opening and dissociation and, therefore, existing in a monomer–trimer equilibrium [50].

The F and G proteins are the only RSV antigens known to induce antibodies that can directly neutralize the virus [49]. However, it is still not known if human polyclonal serum antibodies to G can neutralize RSV. Although antigenic drift has been insufficiently studied for both proteins, the F protein is highly conserved among all known currently circulating RSV genotypes, while the G protein displays higher sequence variability [51,52,53]. Therefore, the primary target in RSV vaccine research is the F protein [54]. The prefusion protein conformation consists of six potential antigenic sites (Ø, I–V) [55]. During the rearrangement of the prefusion conformation into the postfusion conformation, antigenic sites Ø and V are lost [56,57]. Therefore, sites Ø and V are prefusion-specific, while sites II-IV are present on both conformations [58,59,60]. Monoclonal antibodies (mAbs) bind with varying neutralizing potencies to the different antigenic sites, with the highest neutralizing capacities towards the prefusion-specific antigenic sites (Figure 3) [61]. Whether this is also the case for human serum antibodies is not clear.

## 3. History of the Development of RSV Vaccines

The search for RSV treatments and vaccines has been a long road, characterized by several failures [62]. RSV was identified in 1956, and attempts to develop a vaccine were made in the following years [1,63]. The first attempt, in which a formalin-inactivated virus (FI-RSV) was evaluated as a vaccine, was, however, a dramatic failure. Infants immunized with FI-RSV developed worse symptoms than a natural infection, including bronchoconstriction and severe pneumonia. On top of that, two of the immunized toddlers died. The cause of this catastrophe is thought to be the induction of enhanced respiratory disease (ERD), characterized by a non-protective immune response in which antibodies are elicited to target non-protective epitopes [64,65,66]. FI-RSV vaccination, indeed, has shown to elicit a poor neutralizing antibody (nAb) response, probably due to the denaturation of neutralizing antigens associated with a decrease in prefusion F and an increase in postfusion F protein. However, the complete mechanism is still not fully understood [66,67].

After the FI-RSV trial, it became crucial to understand the underlying mechanisms of ERD. As a result, in the following decades, significant research was conducted to clarify the pathogenesis of ERD and deepen the understanding of RSV biology and pathology [1,65]. This has enabled the development of the first prophylactic mAb against RSV, palivizumab (Synagis^®^). This mAb targets antigenic site II of the F protein [68,69,70,71]. It (i) inhibits fusion of the viral particle with the cell membrane and (ii) avoids cell-to-cell spread of the virus, preventing syncytia formation in the lungs [68,72]. This immunoprophylaxis is recommended for infants who are at a heightened risk for RSV due to prematurity or the presence of underlying cardiopulmonary disease [73], with the objective of reducing the number of hospitalizations [74]. However, there are some limitations and disadvantages to this therapy: (i) treatment is very expensive, (ii) it is only administered to high-risk infants, (iii) efficacy is limited and (iv) monthly administration during RSV season is required [27,75,76]. Therefore, the need for further research has persisted in order to generate new treatments and vaccines. Recently, a second mAb, nirsevimab, became available and will be discussed more in detail later.

Over the past 15 years, RSV vaccine research has shifted from experimental to a more rational and structure-based design. An important milestone in the structure-based design of RSV vaccines was the stabilization of F in the prefusion conformation using antibodies and the structural characterization of this prefusion F [77]. The F protein was previously considered a good vaccine antigen because of its necessary function in infection, its high neutralizing properties and its high conservation between viral subtypes, unlike the G protein [51,52,54,78,79]. The prefusion conformation of the protein was found to have some additional immunogenic properties. Several studies showed that nAbs binding exclusively to the prefusion-specific antigenic sites (Ø and V) are much more potent than nAbs binding to antigenic sites located on both conformations [61,80,81]. Prefusion-specific site Ø antibodies, such as 5C4, AM22 and D25, are up to 10- to 100-fold more potently neutralizing compared to palivizumab [58,61]. Based on the prefusion F crystal structure in complex with D25, a stabilized prefusion form of F (Ds-Cav1) was made by introducing specific mutations that lock F in the prefusion state, thereby allowing the conservation of antigenic sites Ø and V [82]. Upon immunization, high neutralizing titers were observed in both mice and rhesus macaques, respectively, up to 8-fold and 80-fold compared to postfusion F immunization. The characterization of prefusion F and the generation of the prefusion-stabilized F construct had an enormous impact on the RSV vaccine field, and many vaccine candidates and recently approved vaccines are based on this finding [78].

## 4. Contemporary RSV Vaccine Landscape with Emphasis on Vaccines’ Constructs

The RSV vaccine landscape currently includes more than 30 vaccine candidates in different stages of development, using different vaccine antigens and technologies. Two candidates were recently approved by the Food and Drug Administration (FDA), namely, GSK’s RSVpreF3 vaccine (Arexvy^®^) and Pfizer’s RSVpreF vaccine (Abrysvo^®^), both protecting the elderly (60+) [83]. Pfizer’s vaccine is also approved for maternal vaccination. While the majority of the RSV vaccines and vaccine candidates are based on prefusion-stabilized F constructs, other antigen designs are also being evaluated. Thus, for the G, SH and N proteins, there is only one vaccine candidate in development at a time. Here, we give an overview of the different antigen designs used for RSV vaccine development, including the recently approved vaccines. A further distinction is made based on the incorporation of one or multiple RSV proteins in the vaccine (Figure 4)

### 4.1. F Protein

The F protein is currently the major target in vaccine design due to its unique properties: (i) most neutralizing epitopes, (ii) critical role in cell entry and (iii) high conservation between different RSV subgroups [51,52,54,78,79]. Although the prefusion-stabilized structure is the most common antigenic construct, others are being investigated, ranging from the full-length F protein to specific antigenic sites of F. Below, the vaccines targeting the F protein will be discussed in more detail.

#### 4.1.1. Full-Length F Protein

**The National Institute of Allergy and Infectious Diseases (NIAID)** utilized a chimeric vaccine with the F protein from RSV as an antigen. To optimize the vaccine, they employed a genome replication-deficient Sendai virus (SeV) vector, which offers various advantages. One key advantage is the absence of preexisting immunity in humans, which minimizes the likelihood of the vaccine being quickly cleared by the immune system. Additionally, the SeV vector efficiently infects ciliated airway epithelial cells, making it suitable for mucosal delivery and stimulating localized immune responses [84,85,86,87,88]. To create the chimeric vector, the SeV F ectodomain was swapped with the RSV F ectodomain. This modification ensured that the antigen was actively expressed on the vaccine particles’ surfaces in its native prefusion conformation, contributing to effective neutralizing responses [56,58,82,84]. Until now, the vaccine has demonstrated the ability to induce both humoral and cellular immune responses in mice. Following intranasal immunization with SeV/RSV, the vaccine successfully cleared RSV from the mice’s lung tissue, while intramuscular immunization also led to a substantial reduction in RSV levels after a live RSV challenge [84]. The vaccine candidate is in clinical phase 1, where it is being evaluated for safety and immunogenicity in healthy adults (NCT03473002) [89].

Additionally, **Blue Lake** is progressing its parainfluenza virus 5 (PIV5)-based vector RSV vaccine, named BLB-201. The attenuated vector is modified in such a way that it expresses the RSV F protein. Preclinical studies already showed the ability to induce humoral and cellular immune responses and protection against RSV challenge infection in several animal models. The vaccine candidate is situated in clinical phase 1/2a, targeting the pediatric population (NCT05655182 and NCT05281263) [90,91,92].

#### 4.1.2. Prefusion-Stabilized Soluble F Protein

As discussed above, the prefusion-stabilized F protein has been widely explored as a vaccine antigen. Currently, most RSV vaccine candidates are relying on a type of prefusion-stabilized F protein because of its capacity to induce high levels of nAbs [51,52,54,78,79]. The main distinctions between the vaccines and vaccine candidates are often subtle differences in stabilizing mutations and the use of different vaccine platforms, like subunit, nucleic acid and recombinant vector vaccines.

**GSK**’s RSVpreF3 vaccine is a protein subunit vaccine based on the prefusion F protein that is combined with GSK’s proprietary AS01E adjuvant [82,93]. The construct is derived from the RSV A2 strain, the prototype RSV lab strain [94,95]. The antigen construct is based on the prefusion-stabilized structure of Prof. Dr. McLellan with the following mutations: the introduction of a disulfide bridge (S155C and S290C) and a hydrophobic cavity (S190F and V207L) [82,96]. This antigen construct was developed for elderly and maternal immunization. In maternal immunization, antibodies generated by the mother are passively transmitted to the infant through the placenta, thereby providing protection against RSV in the first few months of life [97]. Despite the proven efficacy of maternal immunization for other viruses (including influenza and pertussis) [98,99], the RSVpreF3 maternal vaccine study was discontinued in a clinical phase 3 study (GRACE study) (NCT04605159) [100]. The discontinuation was due to an imbalance in the proportions of preterm births and neonatal deaths in the RSVpreF3 group vs. the placebo group. This imbalance was seen mainly in LMICs, where significantly more preterm births were registered. In HICs, the incidence of preterm births was similar to that in the placebo group and not statistically significant. However, further analyses are ongoing to better understand these safety data [101].

The same antigen design was used for a candidate vaccine for the elderly (60+). In this population, excellent outcomes were obtained in terms of efficacy, safety and immunogenicity, resulting in approval. The RSVpreF3 vaccine by GSK targeting the elderly was named Arexvy^®^ (NCT04886596). The risk of people aged 60 and older developing LRTI from RSV severe disease was reduced by 94.1% [102,103].

Besides GSK, **Pfizer** also developed a prefusion-stabilized F antigen design. Pfizer’s approach consisted of screening multiple constructs for stability and immunogenicity, followed by the selection of one construct, construct 847. In this construct, three mutations were introduced, namely a disulfide bridge (T103C-I148C), cavity filling (S190I) using non-polar amino acids and a charge neutralization mutation (D486S), to reduce ionic repulsion or enhance ionic attraction between residues that are proximate to each other at the trimer interface in the prefusion conformation. Construct 847 was most able to elicit consistently high nAb titers in mice and cotton rats, approximately 50 times higher than immunization with postfusion F [104]. Pfizer created a bivalent prefusion vaccine (RSVpreF) starting from construct 847, composed of equal amounts of recombinant RSV prefusion F derived from circulating A and B strains and without the use of an adjuvant [104,105]. The inclusion of aluminum hydroxide (Al(OH)_3_) did not provide any immune-enhancing benefits, and the groups that received vaccines containing Al(OH)_3_ experienced more frequent and severe local reactions [106]. This candidate was recently approved and will target infants via maternal immunization [107,108]. The vaccine was shown to be effective against medically attended severe RSV-associated LRTI within 90 days and 180 days, with 81.8% and 69.4% reduction in LRTI, respectively (MATISSE trial—NCT04424316) [109,110]. In the study, a somewhat higher number of preterm births were detected in the vaccinated group compared to the placebo group; however, this was not statistically significant [101].

The bivalent antigen construct (RSVpreF) was also developed as a vaccine for the elderly and showed high efficacy in the RENOIR trial (NCT05035212) [111]. The vaccine reduced RSV-associated LRTI with >2 or >3 symptoms by 66.7% and 85.7%, respectively [112]. The vaccine has been approved for use in the elderly and is called Abrysvo^®^.

**Moderna** has developed an mRNA vaccine targeting the elderly (60+), mRNA-1345, that is currently in clinical phase 3 (NCT05127434) [113]. This vaccine consists of a single mRNA sequence and encodes for a stabilized prefusion F protein [114]. However, the exact sequence has not been communicated yet. For this vaccine, Moderna uses the same lipid nanoparticle technology as for its COVID-19 vaccine. Topline results were received in the ConquerRSV trial, which resulted in breakthrough designation from the FDA [114]. mRNA-1345 is currently also in clinical phase 1 for pediatric and maternal immunization (NCT05743881) [115].

**Janssen Pharmaceutical** has developed Ad26.RSV.preF, an adenovirus vector-based vaccine that expresses a prefusion-stabilized F antigen, for elderly (60+) (NCT04908683) and pediatric use (seropositive children 12–24 months) (NCT03303625) [116,117]. The antigen construct was designed with three different mutations in the F protein to stabilize the antigen in the prefusion configuration. Firstly, a proline residue was inserted (S215P) to avoid assembling the prefusion refolding region 1 (RR1) structural components into a single, long coiled coil, which occurs upon shifting to the postfusion state. This substitution provides a substantial stabilizing effect. Secondly, a hydrophobic residue (N67I) was introduced that further stabilized prefusion F by increasing hydrophobic interactions. Finally, negative charge repulsion was reduced at the interface between the protomers by an E487Q substitution, which also allowed the formation of hydrogen bonds of glutamine across the trimer with D486 from another protomer. Combined, these mutations led to a novel construct with high production efficiency and stability, which induced protection against RSV in a cotton rat model [78]. Besides antigen design, Janssen Pharmaceuticals also studied the quality of two possible vaccine platforms: PreF as a protein in a subunit protein vaccine and preF in an adenovirus vector (Ad26.RSV.preF). The study revealed that Ad26.RSV.preF provided a qualitative enhancement of immune responses compared with the subunit vaccine in a cotton rat model. Therefore, the combination of Ad26 and prefusion F protein (Ad26.RSV.preF) was chosen for the induction of a more optimal immune response [118,119].

The Ad26.RSV.preF vaccine was in clinical phase 3 for the elderly (60+), but Janssen decided to discontinue it after a portfolio review, even though the construct was very promising [120,121,122]. The same construct in clinical phase 2, being tested in seropositive children with an age of 12–24 months, is also halted [116,123].

Another vaccine used the Ds-Cav1 prefusion-stabilized structure. This subunit protein, called VRC-RSVRGP084-00-VP, targets the elderly (60+) and infants via maternal immunization (NCT03049488) and is funded by the **National Institute of Health** (**NIH), the National Institute of Allergy and Infectious Diseases (NIAID) and the Vaccine Research Center (VRC)**. To prevent the rearrangement into the highly stable postfusion conformation, three modifications were introduced: (i) the addition of the T4-phage fibritin trimerization domain (“foldon”) to the ectodomain, (ii) the introduction of cysteine residues (S155C and S290C) and (iii) cavity-filling hydrophobic substitutions (S190F and V207L) [82,124,125]. The vaccine candidate is in clinical phase 1, where its safety, tolerability and immunogenicity will be evaluated [125].

**Icosavax** has a bivalent experimental vaccine, IVX-A12, designed to target both RSV (IVX-121) and human metapneumovirus (hMPV) (IVX-241) in the elderly. To achieve this, they employed a classic construct, the prefusion-stabilized F conformation, but adopted a different vaccine platform, known as Virus-Like Particle (VLP) technology. This VLP platform enables the display of complex viral antigens in a multivalent, particle-based manner, aiming to elicit broad, strong and long-lasting protection against the targeted viruses [126,127]. In their approach, Icosavax created a self-assembling protein nanoparticle that presents 20 copies of Ds-Cav1 on its exterior surface. The use of this full-valency nanoparticle immunogen resulted in significantly higher nAb responses, around 10-fold greater, compared to the trimeric DS-Cav1, as observed in both mice and non-human primates [128]. Currently, IVX-A12 is in clinical phase 2, where safety and immunogenicity assessments will be conducted (with and without the use of CSL Seqirus’ proprietary adjuvant MF59^®^) to further evaluate its efficacy and potential as a vaccine (NCT05664334) [129].

#### 4.1.3. Prefusogenic F Protein

Besides the prefusion-stabilized F protein, prefusogenic vaccines have also been explored as vaccine antigens. As previously mentioned, F initially exists as an inactive precursor called F0 and is cleaved twice by furin-like proteases during maturation, resulting in the release of the p27 peptide [43]. Exactly when and where this cleavage occurs has been questioned in recent years by various research groups [49]. Whereas initially it was thought that this peptide was not exposed to the immune system and was, therefore, considered irrelevant, there is increasing evidence that p27 may play an important role [130,131,132]. Therefore, further research is necessary and could be of importance in vaccine development. So far, only one company has made an attempt to develop a vaccine candidate that includes this peptide.

**Novavax** has evaluated a near-full-length RSV F antigen design that includes the p27 peptide. They made use of a particle-based technology for the production of their RSV F nanoparticle vaccine targeting the elderly (60+) (NCT02608502) and infants through maternal immunization (ResVax) (NCT02624947) [133,134,135,136]. The RSV F nanoparticle construct relies on prefusogenic F. It contains near full-length RSV F with a mutation of the furin cleavage site proximal to the fusion peptide (KKQKQQ to KKRKRR) and a deletion of 10 aa of the fusion peptide (Phe137-Val146). As a result, the vaccine antigen contains p27, unlike other prefusion F-based vaccines [137,138]. For both target populations, clinical studies advanced to phase 3 yet failed to reach their primary and secondary endpoints and were therefore halted [133,134,135,136,139].

#### 4.1.4. Specific Antigenic Sites of the F Protein

**Virometix** employs VLP technology. Their vaccine candidate, V-306, is engineered using multiple RSV F site II protein mimetics (FsIIms) as an antigenic epitope. This mimetic is conjugated to a specially designed synthetic lipopeptide via an N-terminal amino-oxyacetyl group, resulting in the formation of V-306-SVPL. The lipopeptide within V-306-SVPL comprises several key elements, including a versatile CD4+ T-helper epitope, a stable helical trimer formed by a coiled-coil motif (heptad repeat IEKKIES) and, at the N-terminus, the TLR-2/6 ligand Pam2Cys. Around 60–90 copies of each V-306 lipopeptide chain assemble to form a nanoparticle. The lipid chains are embedded in the core of this micelle-like particle, while the exposed surface showcases the epitope mimetics [140,141,142]. The vaccine has demonstrated the ability to elicit robust, long-lasting and specific immune responses in phase I studies, providing protection against RSV with minimal risk of vaccine-associated ERD (NCT04519073) [141,143]. Additionally, the vaccine construct has undergone preclinical evaluation for intradermal delivery via a needle-free epicutaneous patch, showing promising safety and efficacy in enhancing preexisting immunity against RSV in mice [144].

### 4.2. G Protein

The G protein, also called attachment protein, is a type II integral membrane protein responsible for binding to target cells, utilizing host-cell surface molecules [27,145]. This protein is a non-obvious target for vaccine design as it shows great sequence variability. In fact, it exhibits the most variable regions of all RSV proteins across RSV isolates [146,147]. Moreover, the G protein is not necessary for viral replication in vitro [148,149]. However, a central region in the G protein, called the central conserved domain (CCD), seems to be highly conserved, showing significant antigenic relatedness. This region plays a significant role in functions such as transport, processing, carrying out biological activities and being a target for neutralization in viruses from different subgroups. Additionally, antibodies that target this conserved domain have protective effects in both preventive and postinfection animal models [147,150]. Several other studies also confirm that the RSV G protein generates protective mAbs in vitro and in vivo [150,151,152,153].

One vaccine candidate (to the best of our knowledge) in clinical development is based on the G protein as the target antigen. This construct, named BARS13, was developed by **Advaccine Technology** and targets the elderly and the pediatric population. The structure of the G protein that is employed has not been communicated yet. The vaccine contains the immunosuppressant cyclosporine A in order to induce regulatory T cells [17,154,155]. During clinical phase 1, the candidate was safe and well-tolerated and showed significant levels of antibody response (NCT04851977) [156]. The vaccine candidate is in clinical phase 2, where safety and efficacy will be further evaluated (NCT04681833) [157,158].

### 4.3. SH Protein

The SH protein is a type II transmembrane protein that functions as a viral ion channel [159,160,161]. The protein is known to inhibit inflammasome activation and TNF signaling [159,162]. Unlike the F and G proteins, SH on the virion is poorly accessible to antibodies, and it induces non-neutralizing antibodies, making it unattractive as a vaccine antigen [163,164,165]. However, Schepens et al. described promising results, as the SH protein appears to have an ectodomain (SHe) that is accessible and expressed in infected cells. This SHe is thought to generate non-neutralizing antibodies that utilize antibody-dependent cellular cytotoxicity (ADCC), which activates alveolar macrophages (via FcγRs I and III) to remove RSV-infected cells [165,166,167]. SHe-based vaccination reduces viral replication in cotton rats and mice [164]. These encouraging findings suggest that a vaccination that specifically targets infectious cells can significantly reduce the incidence of acute respiratory infections [167].

**Immunovaccine** created DepoVax (DPX-RSV), which is evaluated in clinical phase 1 (NCT02472548). The vaccine antigen of DPX-RSV is the ectodomain of the RSV-A-SHe protein. It is manufactured on a depot-based lipid-in-oil delivery platform to provide sustained adjuvant and antigen exposure. By using SHe, SHe-specific antibodies are produced, and clearance by alveolar macrophages is initiated as described above. During clinical phase 1, DPX-RSV was tested in adults aged 50–64 years and proved to be safe and effective. Strong immune responses specific to the SHe antigen were observed in the vaccinees [166,167,168,169].

### 4.4. N Protein

The nucleoprotein (N) is a pivotal constituent of the NC responsible for enveloping and safeguarding the RNA genome of RSV [170,171,172]. So far, the N protein has received limited attention as a potential vaccine antigen, in contrast to the more extensively studied F and G proteins [173]. However, it is important to note that the N protein still holds promise despite its relatively understudied status. It has demonstrated potential as a target for generating protective antiviral T cell responses. Notably, the N protein serves as a major carrier of cytotoxic T lymphocyte (CTL) epitopes in humans, making it a primary focus for CTL memory responses [173,174,175,176,177,178,179]. In the context of RSV infection control, protection primarily depends on the presence of nAbs, while the clearance of virus-infected cells requires the coordinated action of natural killer (NK) cells, CD8+ T- lymphocytes and IFN-γ, underscoring the vital role of virus-specific CTLs [173,178,180,181,182,183,184]. Another advantageous characteristic of the N protein is its remarkable conservation across various RSV isolates [175,185,186]. Nevertheless, to date, only a limited number of vaccine candidates have been developed based solely on the N protein [173,185,186,187,188,189,190,191,192]. Further research and exploration of the N protein’s potential in vaccine development are warranted. So far, one company has a vaccine candidate in development that relies on the N protein.

The **Pontificia Universidad Católica de Chile** developed a vaccine candidate that focused solely on N as an antigen, targeting the pediatric population. To achieve this, they utilized recombinant *Mycobacterium bovis* bacillus Calmette–Guérin (BCG) based on the Danish strain 1331, which expresses the N protein [192,193]. This particular vector was chosen because it has the capability to induce Th1 immunity, as it is believed that effective clearance of RSV requires a balanced Th1-type immune response involving the activation of cytotoxic T cells that produce IFN-γ. Moreover, this vector has demonstrated safety in newborns, infants and adults and has been used as a tuberculosis vaccine for decades in several countries [193,194,195,196,197,198,199,200,201,202,203,204,205]. In mouse models, vaccination with rBCG-N-hRSV effectively prevented lung damage associated with RSV, reduced inflammatory cell infiltration in lung tissue and led to early recruitment of CD4+ and CD8+ T cells to the lungs [192,193,206,207]. Furthermore, rBCG-N-hRSV vaccination not only induced protective T cells but also triggered the production of antibodies against various RSV proteins, in addition to the N protein. These antibodies were capable of neutralizing RSV in vitro and belonged to an antibody isotype suitable for clearing the virus through a mechanism known as Linked Recognition [192,193,208]. rBCG-N-hRSV is in clinical phase 1, where it has already demonstrated safety, good tolerability and the ability to elicit an immune response lasting for at least 6 months (NCT03213405) [192,209].

### 4.5. Multiple Viral Proteins and Complete Virus Particles

Another approach to vaccine antigen design is using multiple proteins in the same vaccine construct in order to generate broad protection. This approach has been adopted by several companies.

**Bavarian Nordic** has evaluated a clinical phase 3 candidate that makes use of a viral recombinant platform. Their vaccine candidate employed a poxvirus vector that is modified and non-replicating (Vaccinia Ankara-BN backbone). The antigen design by Bavarian Nordic is different from the other vaccine candidates in clinical phase 3, since it is not solely based on the F protein but also includes the G, M2-1 and N proteins of RSV-A and a second G protein of RSV-B. This resulted in a construct called MVA-BN-RSV (NCT05238025). The advantage of a multicomponent vaccine was confirmed in preclinical mouse models when they were vaccinated with MVA-BN-RSV and subsequently challenged with RSV via the intranasal route. Complete protection against RSV challenge was observed, and this protection was found to rely on the presence of CD4+ and CD8+ T cells, as well as RSV-specific antibodies, including mucosal IgA [9,54,210]. However, MVA-BN-RSV is halted in clinical phase 3 (VANIR trial) since the vaccine candidate did not meet one of its primary objectives of preventing LRTD from RSV [211].

Another company that focuses on multiple RSV proteins is **Meissa**. Their candidate, MV-012-968, is a live attenuated vaccine (LAV) with suppressed or abolished expression of immune suppressors (SH and G proteins) and increased antigen expression of the F protein. By employing synthetic biology, a construct called OE4 was created, which expresses RSV A line 19F and is attenuated by codon deoptimization of the NS1, NS2 and G genes, ablation of the secreted form of G and deletion of the SH gene. Despite significant attenuation in the upper and lower airways of cotton rats, OE4 (RSV-A2-dNS1-dNS2- ΔSH-dG_m_-Gs_null_-line19F) demonstrates high prefusion antigen titers, thermal stability, immunogenicity and effectiveness [212,213]. MV-012-968 is a pediatric intranasal and thus needle-free vaccine that is currently in clinical phase 2, where it will be tested in seronegative children (6 to 36 months) for safety and immunogenicity (NCT04909021) [214,215].

Another LAV currently undergoing clinical phase 1 studies is CodaVax, developed by **Codagenix**, which targets the pediatric population and the elderly [216]. The company utilizes a construct named RSV-MinL4·0. The construct involves introducing four specific mutations into the L protein using codon deoptimization. This process was achieved through repeated stress passaging in vitro and reverse engineering [217]. RSV-MinL4·0 has been tested on African green monkeys (AGMs) and has demonstrated the ability to trigger a strong immune response, both in terms of humoral and cellular immunity. Additionally, the vaccine has proven to protect AGMs from infection when challenged with the wildtype RSV [218]. The vaccine is currently in clinical phase 1, where it is being evaluated for safety, tolerability and immunogenicity (NCT04919109) [219].

**Intravacc** is currently developing an LAV candidate named IT-RSV∆G, which is in clinical phase 1 and targets the pediatric population [220]. The vaccine was created using reverse genetics techniques to construct a recombinant RSV strain that lacks the G gene (ΔG), based on a clinical RSV isolate known as strain 98-25147-X, which belongs to RSV serogroup A. In studies involving cotton rats, a single dose of the IT-RSV∆G vaccine provided long-lasting protection against subsequent RSV challenges and, importantly, did not induce ERD [221].

Furthermore, there are several LAV candidates that initiated clinical phase 1 and 2 studies that were developed by NIAID and/or Sanofi. It is not sure whether these vaccines are already halted or are still in development since no new updates have been published. These LAV vaccines vary from each other by specific deletions, modifications and mutations. These vaccine candidates are listed in Figure 4.

### 4.6. Vaccines with Unknown Antigen Design

Currently, there are three vaccines in development for which the specific antigens in the vaccine design have not been publicly disclosed. Among them, **Dacchi Sankyo** is advancing its vaccine candidate, VN-0200, which is in clinical phase 2 and targets the elderly population. The vaccine incorporates VAGA-9001a as the antigen and an MABH-9002b adjuvant (NCT05547087) [222]. Additionally, **Sanofi** has two vaccine candidates in development, and the antigen construct of both is unknown. One candidate is a LAV that is named SP0125 (VAD00001) and is in clinical phase 2 (NCT04491877) [223]. The other candidate is an RNA-based vaccine designed for the elderly, and it is currently in clinical phase 1 (NCT05639894) [224,225]. The vaccine is called SP0256 and is a combined respiratory vaccine that consists of a ‘backbone’ that includes RSV, human metapneumovirus and parainfluenza virus [226].

## 5. Immunoprophylaxis

Besides vaccines, mAbs for passive immunization are also a way to prevent RSV infection [227]. Palivizumab (Synagis^®^) has been on the market since 1998 [228], but has its limitations, as discussed above. Therefore, the challenge was to identify new mAbs that address these limitations [27,75,76]. Currently, next-generation mAbs are in late clinical phases or approved [229,230].

Nirsevimab (MEDI-8897) is one of these next-generation mAbs [17] and was developed by MedImmune/AstraZeneca and Sanofi. This mAb is available as of this winter season under the brand name Beyfortus^TM^ [9,229]. Nirsevimab, which appears to be a very potent, fully human mAb, is produced from the parental antibody D25, which was generated from human B cells [231]. This antibody targets the highly sensitive antigenic site Ø, exposed to the prefusion F conformation [17]. A crucial modification in the mAb for its improved activity was a three-aa mutation, YTE, in the Fc part of palivizumab. Previously, it was discovered that the IgG and neonatal Fc receptor (FcRn) had a pH-selective interaction that is essential for extending the circulation half-lives of IgG molecules through intracellular trafficking and recycling [232]. This mutation resulted in a significant increase in the half-life of nirsevimab, so that only one injection is required for the entire RSV season (150 days with nirsevimab vs. 19–27 days with palivizumab). Consequently, the cost can be reduced, which results in the expansion of the target group from high-risk to all infants [17,231]. In addition, there is a rapid onset of action as no activation of the immune system is required [233]. This mAb has been specifically developed to safeguard newborns throughout their initial exposure to RSV during the first year of their lives, as well as children up to 24 months old who are at continued risk of experiencing severe RSV-related illnesses during their second RSV season [229].

In addition to nirsevimab, a second next-generation mAb is in development (clinical phase 3) by Merck, namely clesrovimab (MK1654), also for use in young infants (NCT04767373) [230]. This mAb is fully human and directed against the F protein antigenic site IV. By partial targeting of site V, there is preferential binding to the prefusion conformation [17,230]. Several studies have shown that sites III and IV are highly conserved regions of the F protein, decreasing the risk of emerging antibody-resistant viruses [55,234,235]. Like nirsevimab, clesrovimab has been subjected to the same YTE mutation that extended its half-life [231]. The phase 3 clinical study is expected to be completed by the end of April 2026 and will provide us with more information regarding efficacy and safety (NCT04938830) [236].

Three additional mAbs are currently undergoing clinical studies. One of these mAbs, known as RSM01, is in clinical phase 1 and is being developed by the **Gates Medical Research Institute**. RSM01 is designed to target site Ø, which is exposed on the prefusion F conformation of a specific biological target (NCT05118386) [237,238]. A biosimilar for palivizumab, called Narsyn^®^, is being developed through a public–private partnership involving the **Utrecht Center for Affordable Biotherapeutics**. Presently, Narsyn^®^ has advanced to clinical phase 2 of development (NCT04540627) [239]. Lastly, **Trinomab Biotechnology** is developing TNM-001, which targets the RSV F protein and is currently in clinical phase 2 (NCT05630573) [240,241,242].

## 6. Current Challenges

Despite the significant advances made in recent years and the recent availability of vaccines for the elderly, there are no effective RSV vaccines to efficiently protect older infants. A significant challenge in combating RSV lies in the development of a vaccine specifically tailored for children between the ages of 6 months and 5 years old. Presently, most advanced vaccine candidates primarily focus on the elderly population, leaving children in a vulnerable position. Although there are potential solutions for infants, such as maternal vaccines and immunoprophylaxis, these approaches do not fully address the medical need [2,83,243]. Passive transfer of maternal antibodies provides temporary protection for infants, but this safeguard rapidly diminishes after 3 months. Moreover, maternal vaccination may introduce an additional concern by delaying the child’s own immune response to RSV [76,244,245]. Consequently, these preventive measures primarily target infants younger than 6 months, partially mitigating the problem while leaving a substantial portion of the target population (older infants and toddlers) susceptible to infection [2,243]. In the past decade, the development of pediatric RSV vaccines has remained in the early stages, with most candidates not progressing beyond clinical phase 1 or phase 2 [83]. The challenges associated with developing a pediatric vaccine necessitate careful consideration, particularly regarding the prevention of ERD [64,246].

In addition to the challenge of developing vaccines for children, it is crucial to closely monitor the emergence of mutations in RSV. Surveillance studies focusing on RSV A and B subtypes have revealed that RSV exhibits polymorphisms that can undergo evolutionary changes over time [235]. Moreover, there has been insufficient evaluation of how viruses belonging to different genotypes and subtypes elicit varying levels of antibody responses [247]. This variability poses a risk, potentially reducing the efficacy of developed vaccines and therapeutics, particularly those targeting a limited number of epitopes [54,247,248,249]. Immunoprophylaxis strategies like nirsevimab and palivizumab, which target a single specific epitope [17], and vaccines that focus on only a few epitopes, such as prefusion-stabilized F protein vaccines like RSVpreF3 and RSVpreF, which primarily act on site Ø, may be particularly susceptible to this risk [54]. Furthermore, certain antigenic sites demonstrate a higher susceptibility to variability. For example, antigenic sites III and IV of the F protein have shown a high degree of conservation, and it could be reasoned that this makes these antigenic sites more desirable targets with a reduced likelihood of generating antibody-resistant viruses [55,234,235]. The relevance of these antigenic differences between circulating viruses becomes evident when looking at Suptavumab, a human mAb targeting antigenic site V that was recently discontinued from further development in clinical phase 3. Initially, this mAb demonstrated superior neutralization of RSV-A and B compared to palivizumab. However, the spontaneous emergence of two mutations (L172Q and S173L) in the F protein of circulating RSV B strains led to an almost complete lack of neutralizing activity [247,250,251,252]. Through surveillance efforts, mutations in the F protein’s antigenic sites and in major domains like the mucin-like domain and the CCD of the G protein have been observed [247,248,249,253,254].

These examples highlight the significant consequences of mutations and emphasize the need for preventive measures that can withstand them. One approach is to develop vaccines and immunoprophylaxis that consider these variable antigenic sites, e.g., by creating a cocktail of mAbs targeting different epitopes to mitigate antibody resistance rather than relying solely on a single epitope [17,83]. Additionally, using contemporary clinical isolates instead of prototype strains may prove beneficial during vaccine development and evaluation [255]. Significant differences have been observed between RSV prototype strains and contemporary clinical isolates, including variations in viral replication kinetics, thermal stability, fusion capacity and the nAbs they induce. Therefore, utilizing contemporary isolates ensures a more representative assessment of vaccine efficacy against current circulating viruses [256,257,258,259,260,261,262,263]. Moreover, it could be considered highly desirable to include multiple recent isolates of RSV instead of using just one. For example, variations in the sequence of the RSV F protein between RSV A and B clusters in the epitope Ø region are revealed. RSV A and RSV B viruses can prevail in different seasons, with cycles of 2–3 years depending on the geographical location. Hence, considering these strain differences and the cyclic nature of RSV dominance, a multistrain approach could have benefits for an effective RSV vaccine [104,264,265]. Additionally, it could be interesting in the future to use human serum studies to better understand neutralization, antigenic drift and antibody breadth, as well as for antigenic surveillance. Lastly, comprehensive surveillance using not only whole genome sequencing but also clinical virus isolates is vital for the timely detection and analysis of variants and mutations, facilitating proactive measures to address emerging challenges effectively [247,266].

Finally, besides the well-known antigenic sites on the RSV F protein, other novel potential antigenic sites should also be considered. Recent indications for new epitopes of interest could be the flexible breathing state of the prefusion conformation of the F protein. As discussed earlier, the prefusion conformation is not locked in a fixed state like the prefusion-stabilized F antigens but rather in a flexible state that allows dissociation of the monomers, which could thereby expose new epitopes [50]. Furthermore, there are indications that p27 could possibly be of interest, as recent studies suggest complete removal of p27 may not occur during virus maturation but may occur at a later stage, e.g., during entry. Whether such observations are also happening during natural infections or might rather be confined to some laboratory observations in specific cell lines or resulting from the use of specific virus isolates is not clear [49,130,131,132].

## 7. Conclusions

In recent years, remarkable progress has been made in the development of RSV vaccines. Notably, more than 30 vaccine candidates are currently in development, and 2 vaccines targeting the elderly population and the first maternal vaccine have been successfully introduced to the market. Most of these vaccine candidates focus on the prefusion-stabilized F protein of RSV due to its advantageous properties, including highly neutralizing epitopes and significant conservation across RSV A and RSV B subtypes. However, despite these promising advancements, several unresolved issues remain. There is still a substantial unmet need for an effective pediatric vaccine. Maternal vaccines and immunoprophylaxis do not provide a definitive solution for protecting children against RSV. Besides this, ensuring the retained efficacy of developed vaccines and the early detection of mutations in circulating RSV strains is essential. Therefore, the duration of vaccine protection and the potential need for revaccination also warrant further investigation. To address these challenges, it is imperative to broaden the scope of vaccine design beyond a sole focus on vaccines with a locked prefusion F protein structure. Alternative RSV proteins and possible new antigenic sites should therefore be thoroughly explored. Moreover, antigenic surveillance, as seen with the flu, could be useful if vaccines need to be updated to reflect antigenic changes in circulating viruses. While substantial knowledge has been accumulated, several critical issues remain unresolved. Consequently, further research and exploration of antigen design strategies are vital to the successful development of vaccines that can effectively address the current challenges in combating RSV.

## Figures and Tables

**Figure 1 vaccines-12-00097-f001:**
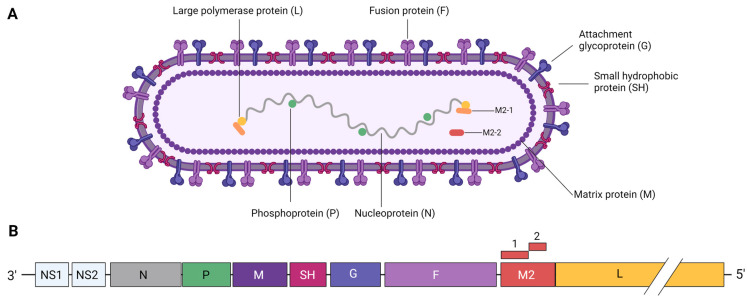
Schematic overview of an RSV virion (**A**) and its genome (**B**). NS1 = non-structural protein 1, NS2 = non-structural protein 2, P = phosphoprotein, M = matrix protein, SH = small hydrophobic protein, G = attachment glycoprotein, F = fusion protein, M2-1 = transcription processivity factor, M2-2 = transcriptional regulatory protein and L = large polymerase protein. F and G are both presented as hypothetical trimers. While F is likely a trimer on the virion, the oligomeric nature of G is still not known. Created with BioRender.com.

**Figure 2 vaccines-12-00097-f002:**
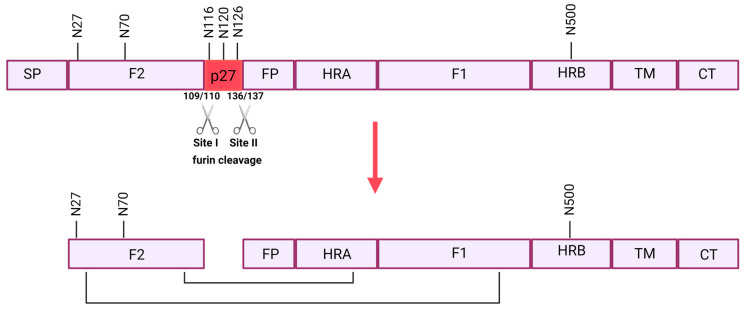
Schematic overview of the maturation process of the F protein. In the top panel, the inactive RSV F0 precursor protein with the different domains, including the two subunits F1 and F2 with the corresponding cleavage sites (I and II), p27, SP = signal peptide, FP = fusion peptide, HRA = heptad repeat A, HRB = heptad repeat B, TM = transmembrane peptide, CT = cytoplasmic. Below, the active F protein is depicted. p27 is removed, and two disulfide bridges are formed between F1 and F2. Six N-glycosylation sites are located at the respective amino acid positions (N27, N70, N116, N120, N126 and N500) [47]. Image created with BioRender.com (2023).

**Figure 3 vaccines-12-00097-f003:**
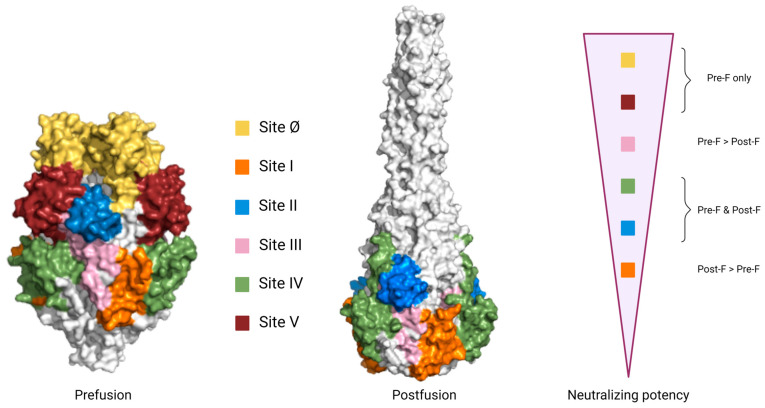
Trimeric prefusion and postfusion conformations of the F protein are shown, in which the different antigenic sites are color-coded. Antigenic sites II and IV are present on both the pre- and postfusion conformations, while sites Ø and V are prefusion-specific. On the right, the relative neutralizing potency of each site is displayed. It should be noted that these differences in potency could also partially reflect differences in antibody affinity or specific properties of the mAbs. Created with PyMOL and BioRender.com.

**Figure 4 vaccines-12-00097-f004:**
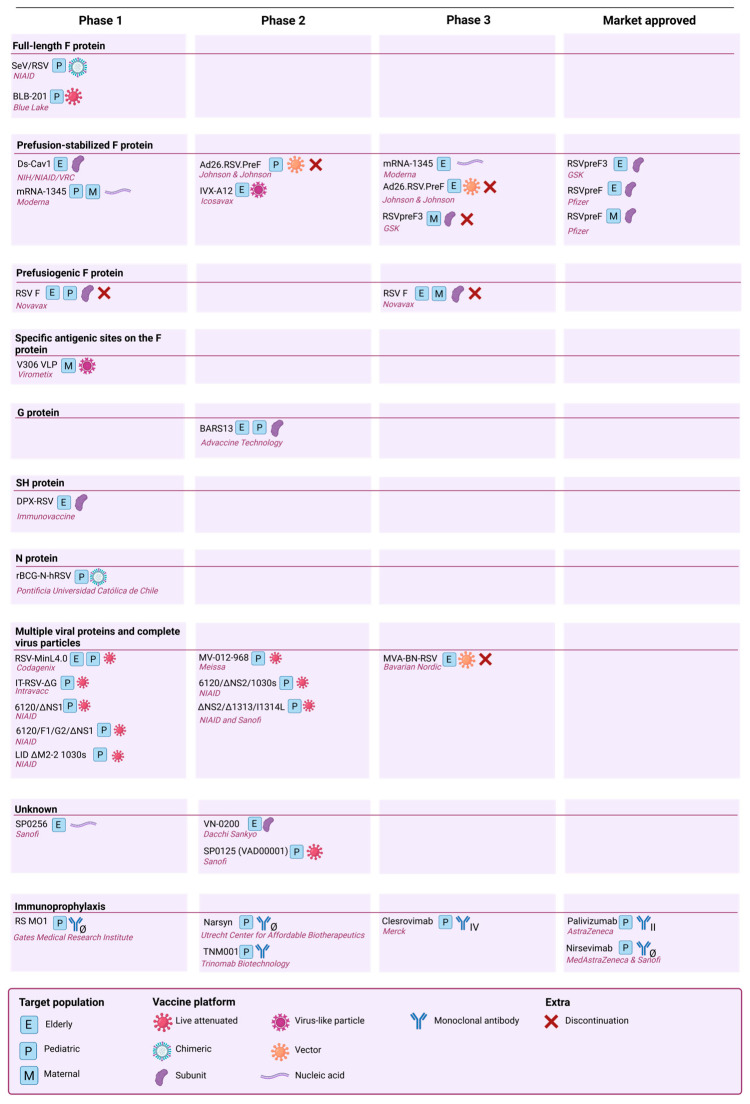
Overview of current vaccines and immunoprophylaxis in development and on the market. Classification according to the antigen construct used in the vaccine. Created with BioRender.com.

## Data Availability

No new data were created or analyzed. Data sharing is not applicable to this article.

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
