# Peer review of "All Eyes on the Prefusion-Stabilized F Construct, but Are We Missing the Potential of Alternative Targets for Respiratory Syncytial Virus Vaccine Design?"

_vaccines, 2024, doi:10.3390/vaccines12010097_

Round 1

Reviewer 1 Report

Comments and Suggestions for Authors

This manuscript provides a more systematic overview of the current status of RSV vaccine design as well as future plans, which is a more comprehensive summary of past research and can also provide a theoretical basis for future research and the development of novel RSV vaccines. I note that the title of this paper focuses more on the prefusion F construct of RSV vaccines, but the full text is not limited to the F protein of RSV, and the title is overly complex, so I suggest revising it.

1. In the Introduction section, the author focuses on the symptoms of RSV infection, and the dangers of RSV. I would suggest to include some overviews of RSV, such as RSV is an RNA virus, Pneumovirus, or even the life cycle of RSV and so on, which are also introductions about RSV. 

2. Line 51 RSV represents a substantial public health concern. should be The infection of RSV represents a substantial public health concern.

3. Line 59-60 RSV is a a member of the family Pneumoviridae, associated with the genus Orthopneumovirus, and was recently reclassified by ICTV to the species Human Orthopneumovirus , can be placed in the previous section.

4. Figure 1 Figure 1A,Figure 1B should also be marked in the text. And, are the images cited from other reference? If so please cite. The figures and tables throughout the manuscript have the similar problems, so please ask the authors to revise them together.

5. Line 76-77 There might be some mistakes about the reference. Similar mistakes were also seen in Line 96, Line 124, Line 142, Line 201, Line 506,

6. The Figure 3 seems not very clear, it is recommended to replace the image with a higher resolution one.

7. The second part of this manuscript focuses on the structure of RSV and details the individual viral proteins of RSV, which seems to be a bit of a departure from the main topic of the manuscript, And, the third part of the development of the RSV vaccine should be presented more. I would recommend condensing the section on RSV structure and expanding on the development of the RSV vaccine.

8. Line 390 mAbs and Line 600 CCD should be given the full name at its first appearance.

9. In section 4 author writed a lot about the different structures of the RSV related to vaccines, I suggest a summary of the subsection after 4.7.

10. Please check that all references are correct.

Comments on the Quality of English Language

I suggest that the English language of these manuscript should be minor edited.

Author Response

Dear reviewer,

We would like to sincerely thank you for your detailed review of our manuscript. Your feedback is greatly appreciated. As we understand that some statements might need some clarification, we have adapted our manuscript according to your comments. Details on how we have addressed these questions are listed below. We hope this explanation answers the questions that were raised.

Kind regards,

Sofie Schaerlaekens and Lotte Jacobs, on behalf of all co-authors.

  • This manuscript provides a more systematic overview of the current status of RSV vaccine design as well as future plans, which is a more comprehensive summary of past research and can also provide a theoretical basis for future research and the development of novel RSV vaccines. I note that the title of this paper focuses more on the prefusion F construct of RSV vaccines, but the full text is not limited to the F protein of RSV, and the title is overly complex, so I suggest revising it.

We would like to thank the reviewer for the effort that was put into making these suggestions. We acknowledge that the original title of this review is maybe too complex. In response, we have simplified the title to enhance readability and understanding. However, in this review, we aim to underscore a crucial point — that the majority of vaccine constructs, both in development and currently available, predominantly target the prefusion-stabilized F protein and that minimal attention has been given to incorporating other proteins in vaccine designs. Therefore, we wanted to drop the focus of prefusion F proteins even in the title.

We changed the title as follows: All Eyes on the Prefusion Stabilized F Construct, but are we missing the Potential of Alternative Targets for RSV Vaccine Design.

  1. In the Introduction section, the author focuses on the symptoms of RSV infection, and the dangers of RSV. I would suggest to include some overviews of RSV, such as RSV is an RNA virus, Pneumovirus, or even the life cycle of RSV and so on, which are also introductions about RSV.

We thank the reviewer for the critical review of the introduction. We agree with the reviewer that the general information about RSV would be better mentioned here, so we have moved a part of section 2 "RSV structure" to the introduction (line 28-33).

  1. Line 51 RSV represents a substantial public health concern. should be The infection of RSV represents a substantial public health concern.

We would like to thank the reviewer for this remark. As we agree that the wording we used was not entirely correct, we have adjusted this sentence in the manuscript:

  • Line 57-58: Overall, RSV infections represent a substantial public health concern, particularly in young children, elderly and vulnerable populations.

  1. Line 59-60 RSV is a a member of the family Pneumoviridae, associated with the genus Orthopneumovirus, and was recently reclassified by ICTV to the species Human Orthopneumovirus , can be placed in the previous section.

We completely agree with this suggestion of the reviewer and would like to thank the reviewer for pointing this out. We have moved this section to the introduction.

  • Line 28-33: RSV is a member of the family Pneumoviridae, associated with the genus Orthopneu-movirus, and was recently reclassified by ICTV to the species Human Orthopneumovirus (hOPV) [3]. It can be divided into two major antigenic subtypes, RSV-A and RSV-B, based on the sequence of the G protein. These two subtypes can co-circulate with pre-dominance alternating every 1-2 years [4], [5].

  1. Figure 1 Figure 1A Figure 1B should also be marked in the text. And, are the images cited from other reference? If so please cite. The figures and tables throughout the manuscript have the similar problems, so please ask the authors to revise them together.

We would like to thank the reviewer for pointing this out. We agree with the reviewer that it makes more sense to refer to the different panels of fig 1 separately, therefore we have changed this in the text.

  • Line 65-67: The RSV genome consists of a linear, negative-sense, single-stranded RNA molecule of around 15 kb, comprising 10 genes that encode 11 structural and non-structural proteins (see Figure 1B) [28].
  • Line 75-77: Three main viral proteins are displayed on the virus envelope: the small hydrophobic (SH) protein, the attachment protein (G), and the fusion (F) protein (see Figure 1A) [37].

We also want to clarify that the figures and table were made with a licensed version of Biorender.com. We have mentioned this in the legenda underneath each figure, in line with the requirements of Biorender.com. We hope this sufficiently answers this remark.

  1. Line 76-77 There might be some mistakes about the reference. Similar mistakes were also seen in Line 96, Line 124, Line 142, Line 201, Line 506.

We sincerely apologize for this mistake and thank the reviewer for pointing this out. All cross-references have been verified and adjusted.

  1. The Figure 3 seems not very clear, it is recommended to replace the image with a higher resolution one.

We thank the reviewer for pointing this out.  We have changed the figure to a higher resolution.

  1. The second part of this manuscript focuses on the structure of RSV and details the individual viral proteins of RSV, which seems to be a bit of a departure from the main topic of the manuscript, And, the third part of the development of the RSV vaccine should be presented more. I would recommend condensing the section on RSV structure and expanding on the development of the RSV vaccine.

We would like to thank the reviewer for this suggestion. We tried to condense section 2 as much as possible. However, we believe this section is essential for this review since we subdivide all vaccine designs by their construct, and, therefore, the protein it is based on. We therefore consider an  overview of the different proteins of RSV is needed. Regarding section 3, as this is already extensively described in other reviews, we selected the most important “highlights” in the history of the development of RSV vaccines. The highlights that are being discussed, provide a background e.g. why most of the vaccines/vaccine candidates are focusing on the stabilized prefusion F construct. As we believe this section is already quite complete, we chose to not expand this section any further.

  1. Line 390 mAbs and Line 600 CCD should be given the full name at its first appearance.

We would like to thank the reviewer for pointing this out. We wrote the full name for both at their first appearances.

  • Line 162: Monoclonal antibodies (mAbs) bind with varying neutralizing potency to the different antigenic sites, with highest neutralizing capacities towards the prefusion specific antigenic sites (Figure 3).
  • Line 414: However, a central region in the G protein, called central conserved domain (CCD), seems to be highly conserved showing significant antigenic relatedness.

  1. In section 4 author writed a lot about the different structures of the RSV related to vaccines, I suggest a summary of the subsection after 4.7.

We thank the reviewer for this suggestion. We understand that this summary would make the review clearer. However, instead of having this overview at the end of section 4, we started with an introduction about this section directly followed with a table that has the same structure as this section. We understand now that this introduction might have been insufficient to provide a good overview. Therefore, we extended this introduction to make it clearer and more understandable for the reader. We hope this sufficiently answers your remark.

  • Line 222-224: While the majority of the RSV vaccines and vaccine candidates are based on prefusion stabilized F constructs, other antigen designs are also being evaluated. For both the G, SH and N proteins, there is only one vaccine candidate in development each time.

  1. Please check that all references are correct.

We thank the reviewer for this comment. All references were checked and corrected where needed.

In attachment you can find the manuscript with its changes.

Reviewer 2 Report

Comments and Suggestions for Authors

Title: Respiratory Syncytial Virus vaccines: all eyes on the prefusion stabilized F construct, but is there more than meets the eye?

The manuscript aims to review the current landscape of the RSV vaccine field. The authors emphasize the prefusion stabilized F construct and note the potential limitations and unknowns of the approach. The authors summaries the many prefusion F construct in clinical trials, including the unique attributes of the design. The authors also discuss other, non-F based, vaccines and vaccines where the exact formulation is not known.

Comments:

·      Some citations not found and instead “ (Error! Reference source not found)” shown. This made it not possible to fully evaluate the manuscript.

·      Figure 1. It might be useful to mention that F is likely a trimer on the virion and the oligomeric nature of G is still not known.

·      Line 85, F may play role in attachment as well via human nucleolin.

·      Figure 3-It should be made clear that the neutralization potency of F was established with a limited number of mAbs (a single one for most sites) and that potency could reflect antibody affinity, or specific properties of the mAbs. Whether this actually holds for polyclonal human serum is still unknown.

·      Line 133, although G specific mAbs have been shown to neutralize RSV, it is still not known if human polyclonal serum antibody to G can neutralize RSV.

·      Line 134, this should be cautioned with the fact that the antigenic drift of neither F nor G has not been measured.

·      Line 151, In the RSV literature, ERD can refer to Enhanced Respiratory Syncytial Virus Disease or Enhance Respiratory Disease.

·      Line 140, needs to be clear neutralization capacity not yet evaluated with human serum

·      Line 586, it might be would worth noting that antigenic changes between subtypes, genotypes, or within epitopes over time has not been evaluated. Unlike flu and sars-cov-2, we do not currently know the antigenic differences between circulating viruses, which could be relevant given the Suptavumab failure.

·      Discussion, human serum studies may be useful to better understand neutralization, antigenic drift, and antibody breadth.

·      Line 644, antigenic surveillance, such as in flu, could also be useful if vaccines need to be updated to reflect antigenic changes in circulating viruses.

Author Response

The manuscript aims to review the current landscape of the RSV vaccine field. The authors emphasize the prefusion stabilized F construct and note the potential limitations and unknowns of the approach. The authors summaries the many prefusion F construct in clinical trials, including the unique attributes of the design. The authors also discuss other, non-F based, vaccines and vaccines where the exact formulation is not known.

Dear reviewer, we would like to sincerely thank you for your detailed review of our manuscript. Your feedback is greatly appreciated. As we understand that some statements might need some clarification, we have adapted our manuscript according to your comments. Details on how we have addressed these questions are listed below. We hope this explanation sufficiently answers the questions that were raised.

Kind regards,

Sofie Schaerlaekens and Lotte Jacobs, on behalf of all co-authors.

Comments:

  1. Some citations not found and instead “ (Error! Reference source not found)” shown. This made it not possible to fully evaluate the manuscript.

We would like to thank the reviewer for pointing this out. We want to apologize for these mistakes. We revised all the cross-references and made sure the references are correctly inserted in the manuscript.

  1. Figure 1. It might be useful to mention that F is likely a trimer on the virion and the oligomeric nature of G is still not known.

We thank the reviewer for this remark. We added the following sentences to the text below the figure.  

  • Line 85-87: F and G are both presented as hypothetical trimers. While F is likely a trimer on the virion, the oligomeric nature of G is still not known.

  1. Line 85, F may play role in attachment as well via human nucleolin.

We appreciate this suggestion by the reviewer, and we inserted this info in the text.

  • Line 90-92: The F protein mediates fusion of the viral membrane with the host cell membrane, and was shown to interact with nucleolin [40].

  1. Figure 3-It should be made clear that the neutralization potency of F was established with a limited number of mAbs (a single one for most sites) and that potency could reflect antibody affinity, or specific properties of the mAbs. Whether this actually holds for polyclonal human serum is still unknown.

We completely agree with this suggestion of the reviewer. It is indeed correct that the potency of F was established with a limited number of mAbs. Therefore we made a remark below the figure where we state that differences in potency could also reflect differences in antibody affinity, or specific properties of the mAbs.

  • Line 150-152: . It should be noted that these differences in potency could also partially reflect differences in antibody affinity, or specific properties of the mAbs.

  1. Line 133, although G specific mAbs have been shown to neutralize RSV, it is still not known if human polyclonal serum antibody to G can neutralize RSV.

We thank the reviewer for the critical review. Therefore, we added a sentence where we mention that it is still not known if human polyclonal serum antibodies to G can neutralize RSV.

  • Line 154-155: However, it is still not known if human polyclonal serum antibodies to G can neutralize RSV.

  1. Line 134, this should be cautioned with the fact that the antigenic drift of neither F nor G has not been measured.

We appreciate this comment. It is indeed correct that antigenic drift of neither F nor G has been measured. Therefore we added this to the text.

  • Line 155-158: Although antigenic drift has been insufficiently studied for both proteins, the F protein is highly conserved among all known currently circulating RSV genotypes, while the G protein displays higher sequence variability.

  1. Line 151, In the RSV literature, ERD can refer to Enhanced Respiratory Syncytial Virus Disease or Enhance Respiratory Disease.

Thank you for the suggestion, we are aware of this. Therefore, we used 'enhanced respiratory disease' as the full name at the first occurrence in the text, ensuring clarity for the reader.

  • Line 173-175: The cause of this catastrophe is thought to be the induction of enhanced respiratory disease (ERD), characterized by a non-protective immune response in which antibodies are elicited that target non-protective epitopes [73]–[75].

  1. Line 140, needs to be clear neutralization capacity not yet evaluated with human serum

We acknowledge that this should indeed be mentioned in the review. Therefore, we inserted a sentence where we mention that this is not yet clear for human antibodies.

  • Line 165: Whether this is also the case for human serum antibodies is not clear.

  1. Line 586, it might be would worth noting that antigenic changes between subtypes, genotypes, or within epitopes over time has not been evaluated. Unlike flu and sars-cov-2, we do not currently know the antigenic differences between circulating viruses, which could be relevant given the Suptavumab failure.

We thank the reviewer for this remark, we have followed this advice and therefore we made some changes in the text where we inserted these suggestions.

  • Line 616-618: Moreover, it has been insufficiently evaluated how viruses belonging to different genotypes and subtypes elicit varying levels of antibody responses [269].
  • Line 627-630: The relevance of these antigenic differences between circulating viruses becomes evident when looking at Suptavumab, a human mAb targeting antigenic site V, was recently discontinued from further development in clinical phase 3.

  1. Discussion, human serum studies may be useful to better understand neutralization, antigenic drift, and antibody breadth.

We agree with this comment and added this to the section as follows: “Additionally, it could be interesting in the future to use human serum studies to better understand neutralization, antigenic drift and antibody breadth as well as for antigenic surveillance.”

  • Line 653-656: Additionally, it could be interesting in the future to use human serum studies to better understand neutralization, antigenic drift and antibody breadth as well as for antigenic surveillance.

  1. Line 644, antigenic surveillance, such as in flu, could also be useful if vaccines need to be updated to reflect antigenic changes in circulating viruses.

It is true that antigenic surveillance is important and should be mentioned here. Therefore, this sentence was inserted in the conclusion.

  • Line 686-687: Besides, antigenic surveillance, as seen with the flu, could be useful if vaccines need to be updated to reflect antigenic changes in circulating viruses.

In attachment you can find the manuscript with the changes that are made.
